# Absolute and relative intensities of solo, free-form dancing in adults: A pilot study

Aston K. McCullough [1,2,3]*

1 Laboratory for the Scientific Study of Dance, Center for Cognitive & Brain Health, Northeastern University, Boston, MA, United States of America, 2 Department of Physical Therapy, Movement & Rehabilitation Sciences, Bouvé College of Health Sciences, Northeastern University, Boston, MA, United States of America, 3 Department of Music, College of Arts, Media & Design, Northeastern University, Boston, MA, United States of America

* a.mccullough@northeastern.edu

**Data Availability Statement:** All relevant data are within the manuscript and its Supporting Information files.

**Funding:** The study was funded was funded in part by an award from the National Endowment for the Arts (Award #: 1879058-38-C-21); Award PI:

## Abstract

### Objectives

Engaging in dance of various styles confers health benefits among adults; however, additional studies on free-form dancing are needed to quantify its intensity and advance research on dance and health. This study characterized the absolute and relative physical activity (PA) intensities of solo, free-form dancing at self-determined moderate and vigorous intensities in adults.

### Method

Participants ($N = 48$) ages 18 to 83 years old, with 0 to 56 years of dance training experience, engaged in 5-minute free-form dance bouts at respectively self-determined moderate and vigorous intensities, both with and without music. Absolute intensity was measured during bouts using indirect calorimetry (metabolic equivalents; METs = $\dot{V}O_2$ ml·kg$^{-1}$·min$^{-1}$/3.5). Relative intensity was measured by ratings of perceived exertion (Borg scale) and heart rate. Linear mixed effects models were used to assess the relationship between absolute and relative intensity metrics and model covariates.

### Results

On average, the relative intensity of self-determined moderate-to-vigorous intensity dancing, with and without music, was 76% of the age-predicted maximal heart rate; 74% of the heart rate reserve (HRR); and 13 on the Borg scale. By measure of %HRR, all adults reached at least a moderate PA intensity across all dance bouts. The mean absolute intensity of self-determined moderate intensity free-form dancing without music was 5.6 METs, and the presence of music ($\beta = 0.6$) and the intention to dance at a vigorous intensity ($\beta = 1.1$) were both significantly positively associated with METs, as participants engaged in free-form dance; METs were significantly inversely associated with age ($\beta = -0.05$) and BMI ($\beta = -0.10$).

McCullough. Dr. McCullough designed and oversaw the study, acquired the funding, collected all data, analyzed the data, specified and ran all statistical models, and wrote and finalized the manuscript.

**Competing interests:** The authors have declared that no competing interests exist.

## Conclusions

When characterized using HRR, engaging in free-form dance at self-determined moderate-to-vigorous PA intensities provided a sufficient stimulus for all adults to reach a moderate PA intensity, which provides evidence that dancing however one wishes at such perceived intensities may support adults in accumulating the recommended weekly dose of ≥150 minutes of moderate intensity PA.

## Introduction

Participating in physical activity (PA) benefits health [1–3], and research shows dancing is a mode of PA behavior that is associated with positive physiological and psychological health status [4–6]. A recent umbrella review has underscored, however, a present dearth of research that elucidates the differential impact that respective dance styles have on key health parameters and outcomes [7]. PA dose, defined as the intensity, frequency, and duration of a given PA exposure, is known to be associated with health outcomes throughout adulthood [8–10], yet little quantitative evidence exists that can be used for establishing the absolute and relative intensities of many familiar dance styles in young to older adults.

The 2018 Physical Activity Guidelines for Americans encourages adults to engage in PA bouts of any length to receive the health-enhancing benefits associated with accumulating a recommended weekly dose of ≥150 minutes of moderate intensity activity [11]. Free-form dancing, or improvised dancing, encompasses creative motor behavior across a wide range of individual expressions [12], and nearly everyone can engage in free-form dance in their own way. To dance as one wants, or feels inspired at any moment, is fundamental to engaging in free-form dance behavior—ostensibly, one can choose to dance however one wishes, most wherever one wants, and with few resources. Therefore, additional research on the intensity of free-form dancing, which requires no concurrent instruction nor prior training, will advance scientific understanding of its intensity and thereby support studies that seek to characterize PA dose when adults are engaged in free-form dancing. To present knowledge, no prior studies on PA intensity have quantified the absolute nor relative intensities of free-form dance behavior in adults [7,13]. Thus, additional research that specifically quantifies the absolute and relative intensities of free-form dance exposures in young to older adults is essential to advancing free-form dance as a potential mode of health-enhancing PA.

Though sparce, the available studies that have aimed to characterize the relative and absolute intensity of dance behavior in adults have focused specifically on codified dance styles such as ballet, ballroom, modern dance, or partnered disco dances [13,14]. These prior reports have collectively suggested that codified dance styles, which are routinely performed using rehearsed motor sequences, are respectively associated with moderate or vigorous activity intensities. Free-form dance behavior, in contrast to codified dance styles, is highly heterogenous in its expression given the individualized, idiosyncratic characteristics of improvisational dancing and its general lack of rehearsed motor sequences [12]. Prior studies reveal that codified dance styles appear to differ from each other with respect to PA intensity [15–19], and additional studies that quantify the intensity of free-form dance are needed to understand if engaging in unstructured, improvised dance behavior is sufficient for eliciting either moderate or vigorous PA intensities in adults. Moreover, studies show that exposure to music [20,21], and ultra-low frequency sounds [22], may affect motor activity patterns in adults. Though studies have previously reported that gait parameters [21], overall movement volume [22], and

PA intensity [23] may be influenced by the presence of music, the relationship between music exposures and PA intensity while dancing is not presently well-understood. To extend the available literature on music and PA behavior, which shows that playing familiar music may increase motor activity [24] and PA intensity while walking [23], additional studies on music and spontaneous, complex motor behaviors, such as free-form dance, are needed.

The physiological and psychological health benefits associated with engaging in structured, codified, or social dance experiences are apparent [4–6]. Recent studies have begun to demonstrate health benefits associated with engaging in unstructured or free-form dance [25,26]; however, the PA intensity of free-form dance remains generally unknown. Engaging in moderate intensity PA of any bout duration for a total of at least 150 minutes throughout the week is recommended for adults, and additional research is needed on the PA intensity of free-form dancing among adults to determine if dancing however one wishes may be a mode of PA that can elicit at least a moderate activity intensity. Given the inarguably accessible nature of free-form dance, and its growing use in health-related research on the effects of dance, the need to better understand PA intensity during free-form dancing among adults is critical. Therefore, the purpose of this pilot study was to characterize the absolute and relative intensities of solo, free-form dancing, in the presence and absence of music, at self-determined moderate and vigorous intensities among adults.

## Materials & methods

### Sample

Community-dwelling adults ages 18 to 85 years old who were able to walk on a treadmill and engage in moderate-to-vigorous dancing were recruited via flyer, email, and word of mouth. The Physical Activity Readiness Questionnaire for Everyone (PAR-Q+) [27] was used to screen participants for contraindications to moderate-to-vigorous intensity exercise, and the Modifiable Activity Questionnaire (MAQ) [28] was used to assess habitual PA levels over the past three months prior to the time of screening. Prospective participants who reported "yes" to any question on the PAR-Q+ or who reported engaging in less than 90 minutes of moderate-to-vigorous PA (MVPA) per week over the last 3 months on the MAQ were excluded. This study additionally excluded individuals with a pacemaker or other implanted medical device, who smoked cigarettes or used other tobacco products at the time of screening, who were pregnant or trying to become pregnant, who habitually bit their fingernails, or used a corticosteroid medication administered topically, orally, or by injection. All research participants ($N$ = 66) provided informed consent, and the study was approved by the Institutional Review Board (Protocol #2070) at the University of Massachusetts Amherst. Participants were recruited for the study, and data were collected, between 15 October 2021 and 18 August 2023.

### Procedures

All data were collected indoors within a laboratory. After providing informed consent, each participant completed a self-report sociodemographic questionnaire that was administered via REDCap [29]. Participants were also asked to report their total years of professional dance training. Height and body mass were measured in the lab.

On a separate day, participants returned to the lab to complete a series of resting measures including a resting 12-lead ECG and an assessment of their resting metabolic rate. The lab was heated to 23.3 C at least two hours prior to each participant's arrival. Participants arrived in the lab after abstaining from: consuming food or beverages, except water, for 8hrs; consuming caffeine, alcohol, smoking or vaping for 24hrs; engaging in vigorous PA for 48hrs. Resting measures were taken at least 45 minutes after participants had been settled in the lab. Each

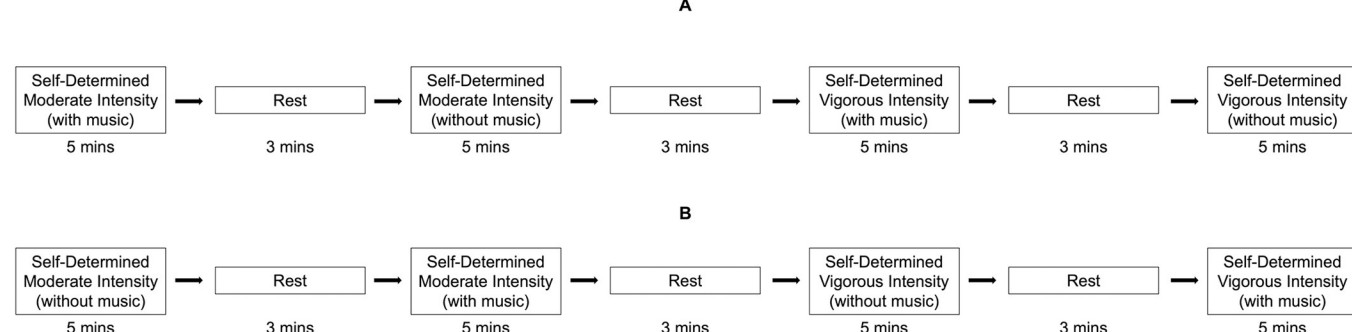

**Fig 1. Sequence of study activities for the self-determined moderate and vigorous free-form dance bouts.** Note: Fig 1A shows the series of study activities completed among adults who engaged in the music-to-no-music free-form dance sequence. Fig 1B shows the series of study activities completed among adults who engaged in the no-music-to-music free-form dance sequence.

participant was fitted with a mask that was connected to a calibrated indirect calorimeter, and ten ECG electrodes were applied to each participant's torso. After being connected to the indirect calorimeter and prepared for the resting ECG, each participant laid supine on a massage table in the lab for 30 minutes with the lights dimmed. At the end of the lab visit, each participant was asked to choose which pieces of music they would use to accompany their dancing during the moderate and vigorous intensity dance bouts. Participants were asked to select pieces of music from their own personal music library; some participants chose to bring only one piece of music to accompany them across dance bouts.

Participants returned to the lab to complete the free-form dance bouts on a separate day. The lab was cooled to 21.1 C at least two hours prior to each participant's arrival. Participants arrived in the lab after abstaining from: consuming food or beverages, except water, for 4hrs; consuming caffeine, alcohol, smoking or vaping for 24hrs; engaging in vigorous PA for 48hrs. Participants were invited to "dance in their usual fashion" both with and without music for five-minute bouts at respectively self-determined moderate and vigorous activity intensities (see Fig 1). During respective moderate and vigorous intensity bouts, each participant was asked to dance at an intensity that they perceived was moderate or vigorous. Participants danced in the laboratory on a 6 m x 5.5 m sprung wood floor covered with marley, and each participant danced in their preferred footwear, in socks, or barefoot. The order of the music exposure for each respective free-form dance bout was assigned randomly (i.e., either "music-to-no music" OR "no music-to-music"). For example, a participant might have been randomly assigned to dance without music playing at a moderate intensity, and then to dance again at a moderate intensity with music playing. Therefore, each participant danced at a moderate intensity twice (once in the presence of music, and once in the absence of music) and then at a vigorous intensity twice (again, in the presence and absence of music). Participants were invited to set the music volume to meet their preference for all selected pieces of music before beginning the first dance bout. Before participants began dancing, the researcher informed each participant that they "would not really be watching" the participant dance and that they would be "mostly focused on the data" during the dance bouts.

Oxygen uptake was measured continuously throughout each dance bout using a portable indirect calorimeter and heart rate was recorded using a chest-worn wireless monitor. At the end of the fourth and fifth minutes of each dance bout, participants were asked to report a rating of their level of perceived exertion. Between each 5-minute dance bout, participants were afforded a 3-minute rest period. If the observed heart rate during any dance bout exceeded

84% of a participant's age-predicted maximal heart rate, a member of the study team asked the participant "to do a little bit less" as the participant continued to dance.

A total of $n = 49$ participants returned to the laboratory and completed the free-form dance session. Device malfunctions were observed while recording oxygen uptake data for $n = 3$ participants; however, all participants except for one were able to stop and repeat the free-form dance trial during which the device malfunction was observed. The single participant who was unable to stop and repeat the trial during which the device malfunction was observed had limited time in the laboratory and was unable to return to the laboratory to repeat assessments. Therefore, only data from $n = 48$ participants with oxygen uptake data were included in analyses. Of the participants who provided informed consent and did not return to dance in the laboratory ($n = 17$), most participant withdrawals were due to a change in participant availability (e.g., the participant became ill, retired, or moved) and was subsequently unable to reschedule ($n = 11$), while the remaining participants withdrew at the request of the researcher ($n = 4$) when there was a change in a participant's health status that affected one's eligibility (e.g., the participant became injured while engaged in non-study related activities outside of the laboratory; or the participant was observed to have a high resting blood pressure while in the laboratory), or no reason for withdrawing from the study was provided by the participant ($n = 2$).

## Measures

**Sociodemographic characteristics.** A self-report questionnaire with items on age (in years), biological sex, gender, race, and ethnicity was administered to participants via REDCap. Self-reported total years of dance training experience was additionally collected.

**Body Mass Index (BMI).** Body mass was measured to the nearest 0.1 kilogram (kg) and height was measured to the nearest 0.1 centimeter using a calibrated seca 286 stadiometer and scale (seca, Hamburg, Germany). Both body mass and height were measured twice, and the mean of each was calculated. Height and body mass were then used to calculate body mass index (BMI) as $kg / m^2$.

**Indirect calorimetry.** A MetaMax 3B-R2 (CORTEX Biophysik GmbH, Leipzig, Germany) portable indirect calorimeter was used to continuously measure $\dot{V}_2$ and $\dot{V}CO_2$ breath-by-breath during resting metabolic rate assessments and during each dance bout. The indirect calorimeter was calibrated to the manufacturer's specifications before testing. First, the device was allowed to initialize for at least 20 minutes. Following, the gas analyzer was calibrated using atmospheric air, and then using a standard gas mixture of 5% $CO_2$ and 15% $O_2$. Disposable turbines, which do not require additional calibration for use with the flow sensor, were used for all tests. Finally, within 15 minutes of beginning any test, a sensor adjustment was conducted using atmospheric air. Raw breath-by-breath data were exported from the CORTEX MetaSoft Studio software for further analyses. Prior to $\dot{V}O_2$ and $\dot{V}CO_2$ analyses, a 15-breath moving average filter, centered on the eighth breath [30], was respectively applied to raw breath-by-breath signals using MATLAB [31].

To calculate resting metabolic rate, the initial five minutes of each breath-by-breath time series was discarded. Steady state epochs were defined as 5-minute windows during which the coefficient of variation (CV) was <10% for both $\dot{V}O_2$ and $\dot{V}CO_2$ signals [32]. A search for the epoch with the lowest CV for both $\dot{V}O_2$ and $\dot{V}CO_2$ was conducted for each participant. Mean $\dot{V}O_2$ and $\dot{V}CO_2$ were then calculated from the selected 5-minute steady state epoch, and RMR was calculated as $\dot{V}O_2$ ml•kg$^{-1}$•min$^{-1}$. Resting energy expenditure data were available for $n = 47$ participants, and data were unavailable for one participant who was unable to complete the test. The mean (standard deviation) within-subject CV for the RMR respiratory quotient ($\dot{V}CO_2/\dot{V}O_2$), as calculated from the selected steady state epochs, was 1.9%(1.1).

To specifically analyze $\dot{V}O_2$ while participants were engaged in steady state PA, [33] $\dot{V}O_2$ data from the final two minutes of each 5-minute dance bout were extracted. The steady state $\dot{V}O_2$ data were converted into an absolute measure of intensity, metabolic equivalents (METs) [13], using $\dot{V}O_2$ ml•kg$^{-1}$•min$^{-1}$ / 3.5 ml•kg$^{-1}$•min$^{-1}$, and the work rate-to-resting metabolic rate ratio (WR:RMR) [34] was calculated as a measure of relative intensity using WR:RMR = $\dot{V}O_2$ ml•kg$^{-1}$•min$^{-1}$ / RMR. METs were classified into PA intensities using the following cut points [13,34]—light ($\geq$1.5, <3), moderate ($\geq$3, <6), and vigorous ($\geq$6).

**Heart rate (HR).** Resting HR (HR$_{rest}$) was measured using a 1200W–Digital RF Wireless 12-lead ECG (Norav, Delray, FL) and the Norav Resting ECG software. A Polar H10 wireless heart rate monitor (Polar Electro Inc., Bethpage, NY) was used to continuously measure HR during dance bouts. Age-predicted maximal HR (HR$_{max}$) was calculated in beats per minute (bpm) for each participant using 207 –(0.7 * age) [35,36].

Heart rate reserve (HRR) was calculated as HR$_{max}$–HR$_{rest}$, and the observed HR during the last two minutes of each dance bout (HR$_{dance}$) were converted into percent of heart rate reserve (%HRR) for each participant using %HRR = (HR$_{dance}$—HR$_{rest}$)/HRR [37]. To classify %HRR into PA intensities, the following cut points were used [38]—light (<40%), moderate ($\geq$40%, <60%), vigorous ($\geq$60%). Observed HR during dance bouts were also converted into percent of HR$_{max}$ (%HR$_{max}$ = HR$_{dance}$/HR$_{max}$) for each participant, and %HR$_{max}$ was classified into PA intensities using the following cut points [38]—light (<64%), moderate ($\geq$64%, <77%), vigorous ($\geq$77%). Both %HRR and %HR$_{max}$ were used as measures of relative intensity during dance.

**Rating of perceived exertion (RPE).** The Borg Rating of Perceived Exertion scale [39] was used to assess perceived exertion on a scale of 6–20 during dance bouts. RPE served as a self-reported measure of relative intensity during dance, and RPE was classified into PA intensities using the following cut points [36,40]—light (<12), moderate ($\geq$12, <14), vigorous ($\geq$14).

**Music tempo.** The Music Information Retrieval Toolbox [41] was used to extract the tempo of each self-selected musical piece that participants chose to accompany their dance bouts.

## Statistical analyses

Data were analyzed in MATLAB [31]. Descriptive statistics are reported as percentages (frequencies), mean (standard deviation), or median (interquartile range).

To test for equivalence between music tempi in the self-selected moderate and vigorous intensity bouts, a non-parametric two one-sided tests of equivalence (TOST) model was implemented. The non-parametric TOST model may be used to simultaneously test for similarities and differences between two variables at selected confidence intervals and equivalence thresholds ($\Delta$) [42]. Model 1a was tested with 95% C.I. and the lower and upper TOST equivalence thresholds $\Delta$ = [–10,10] bpm.

Prior studies have suggested that absolute intensity based upon METs may be affected by differences in resting energy expenditure, given that METs assumes a resting energy expenditure rate of 3.5 ml•kg$^{-1}$•min$^{-1}$. Furthermore, prior research suggests a relative measure of energy expenditure, such as the WR:RMR, may be used as a measure of PA intensity that accounts for individual variability in resting energy expenditure.[28] To test for equivalence between WR:RMR and METs during free-form dance behavior, a TOST was run with 95% C.I. with the lower and upper TOST equivalence thresholds set at $\Delta$ = [-0.75, 0.75] METs. Because some WR:RMR data were found to be non-normally distributed, a nonparametric TOST procedure was implemented (Model 1b).

Linear mixed effects models [43] were used to estimate associations between the relative and absolute intensity of free-form dance bouts, using each respective measure of PA intensity, self-determined intensity (intensity condition), and sociodemographic (age), anthropometric (BMI), and environmental factors (presence of music). All models were run as random intercepts models, adjusting for nested observations within subjects; categorical variables were dummy coded and continuous covariates were grand mean centered. Models 2a – 5a, respectively, are intercept-only models that estimate the relative (models 2–4) and absolute (model 5) intensity of self-determined moderate-to-vigorous intensity free-form dance behavior for %$HR_{max}$ (2a), %HRR (3a), RPE (4a), and METs (5a). Models 2b – 4b, respectively adjust relative intensity estimates for the presence of music (dummy coded—reference category: no music), self-determined intensity (dummy coded—reference category: self-determined moderate intensity), and a music x intensity condition interaction. Model 5b adjusts absolute intensity estimates for age, BMI, the presence of music, self-determined intensity, and a music x intensity condition interaction. Models were run using the residual maximum likelihood [43], with degrees of freedom calculated using Satterthwaite approximations for small sample sizes [44]. Model residuals were all found to be normally distributed, and the assumption of homogeneity of variance was found to be tenable [43]. Beta coefficients (ß) are each presented with their standard error (SE). The significance level was established *a priori* at $\alpha = 0.05$ for all models.

## Results

Table 1 shows sociodemographic, anthropometric, and resting metabolic characteristics for $n = 48$ participants ages 18 to 83 years old, with a range of 0 to 56 years of dance training experience.

The median music tempo during the self-determined moderate intensity dance bouts in the sample was 122(30) beats per minute, and the median tempo of the music during the self-determined vigorous intensity dance bouts was 124(37) bpm. The median difference in music tempi between the self-determined moderate and vigorous intensity dance bouts with music was ~0 bpm (min: -76, max: 120), and the TOST results showed the music tempi were statistically equivalent and not different (95% C.I., -9.9 to 5.5 bpm; $p = 0.036$) during the respectively self-determined moderate and vigorous intensity dance bouts.

### Relative and absolute intensities of free-form dance

Results for each covariate-adjusted linear mixed effects model appear below; Table 2 shows the average relative and absolute intensities of solo, free-form dancing both with and without music.

**Percent of age-predicted maximal heart rate.** Model 2a showed the %$HR_{max}$ reached during self-determined moderate-to-vigorous intensity free-form dancing was 75.5% (95% C.I., 72.3 to 78.6), on average, after adjusting for all other model covariates. The average %$HR_{max}$ during self-determined moderate intensity dancing without music was 68.8% (95% C.I., 65.3 to 72.3) controlling for all other model covariates (Model 2b), and %$HR_{max}$ was significantly higher (4.0% [1.13]; 95% C.I., 1.80 to 6.25; $t_{141} = 3.57$; $p < 0.001$) during dance bouts with music than without, after adjusting for all other model covariates. The average %$HR_{max}$ during self-determined vigorous intensity dancing was significantly higher (9.2% [1.13]; 95% C.I. 6.94 to 11.40; $t_{141} = 8.13$; $p < 0.001$) than during self-determined moderate intensity dancing, after controlling for other model covariates, and there was no significant interaction (0.25[1.60]; 95% C.I. -2.90 to 3.40; $t_{141} = 0.16$; $p > 0.05$) between the music and self-determined intensity conditions (Adjusted $R^2 = 0.85$).

**Table 1. Sociodemographic, anthropometric, and resting metabolic characteristics among community-dwelling young to older adults (n = 48).**

|  | M(SD) |
|---|---|
| **Age** (years) | 42(18) |
| **BMI** (kg/m$^2$) | 24.7(4.4) |
| **Resting HR** (bpm) | 57(7) |
| **Resting Metabolic Rate** $\dot{V}O_2$ ml•kg$^{-1}$•min$^{-1}$/3.5 (n = 47) | 3.5(0.6) |
|  | **%(n)** |
| **Sex** |  |
| Female | 79.2(38) |
| Male | 20.8(10) |
| **Gender** |  |
| Cisgender Woman | 72.9(35) |
| Cisgender Man | 20.8(10) |
| Other gender identity | 6.3(3) |
| **Race** |  |
| White | 70.8(34) |
| Black | 6.2(3) |
| Asian | 8.3(4) |
| Mixed Race | 10.4(5) |
| Other/Not specified | 4.2(2) |
| **Ethnicity** |  |
| Latinx/Hispanic | 14.6(7) |
| Not Latinx/Hispanic | 85.4(41) |

Abbreviations: Mean (M), standard deviation (SD), body mass index (BMI), kilogram (kg), meter (m), beats per minute (bpm), ml (milliliter), minute (min).

**Table 2. Covariate-adjusted relative and absolute intensities of solo, free-form dancing, in the presence and absence of music, among 18- to 83-year-old adults.**

|  | Self-determined Moderate Intensity | | Self-Determined Vigorous Intensity | |
|---|---|---|---|---|
|  | **Music** | **No Music** | **Music** | **No Music** |
| **%HR$_{max}$** (n = 48) | 72.8 [69.3, 76.3] | 68.8 [65.3, 72.3] | 82.2 [78.8, 85.7] | 78.0 [74.5, 81.4] |
| **%HRR** (n = 48) | 71.5 [68.1, 75.0] | 67.3 [63.8, 70.8] | 81.7 [78.2, 85.1] | 76.9 [73.4, 80.4] |
| **RPE** (n = 48) | 11.9 [11.2, 12.7] | 11.8 [11.0, 12.5] | 13.8 [13.1, 14.6] | 13.5 [12.8, 14.2] |
| **METs** (n = 48) | 6.2 [5.7, 6.7] | 5.6 [5.2, 6.1] | 7.5 [7.0, 8.0] | 6.7 [6.2, 7.2] |

Note. Table 2 shows the average [95% Confidence Intervals] relative (%HR$_{max}$, %HRR, RPE) and absolute (METs) intensities of solo, free-form dancing among community-dwelling adults from respective linear mixed effects models. Abbreviations: Percent of age-predicted maximal heart rate (%HR$_{max}$), percent of heart rate reserve (%HRR), rating of perceived exertion (RPE), metabolic equivalents (METs).

**Percent of heart rate reserve.**    Model 3a showed the %HRR reached during self-determined moderate-to-vigorous intensity free-form dancing was 74.3% (95% C.I., 71.2 to 77.5), on average, after adjusting for all other model covariates. The average %HRR during self-determined moderate intensity dancing without music was 67.3% (95% C.I., 63.8 to 70.8) controlling for all other model covariates (Model 3b), and %HRR was significantly higher (4.3% [1.19]; 95% C.I. 1.9 to 6.7; $t_{141}$ = 3.58; $p <$ 0.001) during dance bouts with music than without music, holding all other model covariates constant. The average %HRR during self-determined vigorous intensity dance bouts was significantly higher (9.7% [1.19]; 95% C.I. 7.3 to 12.0; $t_{141}$ = 8.04; $p <$ 0.001) than during self-determined moderate intensity dance bouts, after controlling for all other model covariates, and there was no significant interaction (0.46[1.70]; 95% C.I. -2.9 to 3.8; $t_{141}$ = 0.27; $p >$ 0.05) between the music and self-determined intensity conditions (Adjusted $R^2$ = 0.83).

**Rating of perceived exertion.**    Model 4a showed RPE during self-determined moderate-to-vigorous intensity free-form dancing was 12.8 (95% C.I., 12.1 to 13.4), on average, after adjusting for all other model covariates. Model 4b showed the average RPE during self-determined moderate intensity dancing without music was 11.8 (95% C.I., 11.0 to 12.5) controlling for all other model covariates, and RPE was not significantly higher (0.17[0.21]; 95% C.I. -0.25 to 0.58; $t_{141}$ = 0.79; $p >$ 0.05) during dance bouts with music than those without music, after adjusting for all other model covariates. The average RPE during self-determined vigorous intensity dancing was significantly higher (1.71[0.21]; 95% C.I. 1.29 to 2.12; $t_{141}$ = 8.11; $p <$ 0.001) than during self-determined moderate intensity dancing, after controlling for all other model covariates, and there was no significant interaction (0.19[0.30]; 95% C.I. -0.40 to 0.78; $t_{141}$ = 0.70; $p >$ 0.05) between the music and self-determined intensity conditions (Adjusted $R^2$ = 0.88).

**Metabolic equivalents.**    Model 5a showed that the absolute intensity of self-determined moderate-to-vigorous intensity free-form dancing was 6.5 METs (95% C.I., 6.1 to 7.0), on average, after adjusting for all other model covariates.

Table 3 (Model 5b) shows covariate-adjusted MET values during free-form dancing with and without music at self-determined moderate and vigorous intensities (Adjusted $R^2$ = 0.75), and Fig 2 shows covariate-adjusted METs conditional on age and BMI, respectively, using the fitted results from Model 5b. MET values were significantly higher ($p <$ 0.05) during dance bouts with music than without music, after adjusting for all other model covariates. MET values during self-determined vigorous intensity dance bouts were also significantly higher ($p <$ 0.001) than during self-determined moderate intensity dance bouts. A 20-year difference in age either above or below the sample mean (i.e., 42.3 years) was associated with a respective -0.90 or +0.90 average change in METs (Fig 2A and 2B), holding all other variables constant. A 5-unit difference in BMI either above or below the mean BMI in the sample (i.e., 24.7 kg/m$^2$) was associated with a respective -0.51 or +0.51 average change in METs (Fig 2C and 2D), holding all other model covariates constant.

**Work rate-to-resting metabolic rate ratio & METs.**    The median WR:RMR and METs for adults ($n$ = 47) during self-determined moderate intensity dancing with music were 5.8 (2.3) and 5.8(1.9), respectively, and the median difference between WR:RMR and METs was 0.08 (min: -2.4, max: 2.5). TOST results (Fig 3) showed the WR:RMR and METs were significantly equivalent and not different (95% C.I., -0.64 to 0.44; $p$ = 0.016) during self-determined moderate intensity dancing with music. The median WR:RMR and METs during self-determined moderate intensity dancing without music were 5.4(2.1) and 5.4(2.3), respectively, and the median difference between WR:RMR and METs was 0.11 (min: -1.9, max: 1.9). TOST results showed WR:RMR and METs were significantly equivalent and not different (95% C.I., -0.62 to 0.42; $p$ = 0.009) during self-determined moderate intensity dancing without music.

**Table 3. Absolute intensity of free-form dance behavior at self-determined moderate and vigorous intensities in young to older adults, with adjustments for age and BMI.**

|  | *ß (95% C.I.)* | SE(*ß*) | *t, df* [a] | p-value |
|---|---|---|---|---|
| Intercept [b] | 5.63 (5.16 to 6.11) | 0.24 | 23.13, 85.07 | *p* < 0.001 |
| Age [c] | -0.05 (-0.07 to -0.02) | 0.01 | -4.04, 45 | *p* < 0.001 |
| BMI [d] | -0.10 (-0.19 to -0.01) | 0.05 | -2.20, 45 | *p* = 0.033 |
| Music | 0.59 (0.16 to 1.01) | 0.21 | 2.71, 141 | *p* = 0.008 |
| S-D Vigorous Intensity | 1.09 (0.66 to 1.51) | 0.22 | 5.01, 141 | *p* < 0.001 |
| Music x Intensity Condition | 0.21 (-0.40 to 0.81) | 0.31 | 0.67, 141 | *p* > 0.05 |

Note: Table 3 shows results from a linear mixed effects model on the absolute intensity (metabolic equivalents, METs) of free-form dance behavior among 18- to 83-year-old adults (*n* = 48) in relation to age (years), BMI (kg/m$^2$), the presence of music, and the intention to dance at a moderate or vigorous intensity. Abbreviations: Body mass index (BMI), self-determined (S-D), beta coefficients (*ß*), 95% confidence intervals (95% C.I.), standard error (SE), degrees of freedom (df), kilogram (kg), meter (m).

[a] The linear mixed effects model was fit using the Residual Maximum Likelihood, and degrees of freedom were calculated using Satterthwaite approximations.

[b] Self-determined moderate intensity dancing without music (reference category).

[c] Age was centered at the grand mean (i.e., 42.3).

[d] BMI was centered at the grand mean (i.e., 24.7).

The median WR:RMR and METs during self-determined vigorous intensity dancing with music were 7.0(3.3) and 7.5(2.5), respectively, and the median difference between WR:RMR and METs was -0.09 (min: -3.9, max: 2.6). TOST results showed WR:RMR and METs were not equivalent and not different (95% C.I., -0.80 to 0.50; *p* > 0.05) during self-determined vigorous intensity dancing with music. The median WR:RMR and METs during self-determined vigorous intensity dancing without music were 6.7(2.6) and 6.4(2.4), respectively, and the median difference between WR:RMR and METs was -0.08 (min: -2.5, max: 2.6). TOST results showed WR:RMR and METs were statistically equivalent and not different (95% C.I., -0.57 to 0.63; *p* = 0.049) during self-determined vigorous intensity dancing without music.

## Free-form dance activity intensity classification

**Self-determined moderate intensity.** During the self-determined moderate intensity dance bouts completed with and without music (Fig 4A and 4B), 100%(48) of participants engaged in MVPA as classified by %HRR. RPE values showed 65%(31) of participants perceived they were engaged in MVPA when dancing with music at a self-determined moderate intensity, and 69%(33) perceived they were engaged in MVPA when dancing without music. METs classified 98%(47) of participants as engaged in MVPA when dancing with music and without music, while %HR$_{max}$ classified 79%(38) in MVPA when dancing with music and 71% (34) when dancing without music.

**Self-determined vigorous intensity.** During the self-determined vigorous intensity dance bouts completed with and without music (Fig 4C and 4D), 92%(44) of participants engaged in vigorous intensity dancing with music as classified by %HRR, and 94%(45) of participants were classified as engaged in vigorous PA when completing bouts without music. Self-reported RPE values showed 63%(30) of participants perceived they were engaged in vigorous PA while dancing with music, and 58%(28) perceived they were engaged in vigorous PA when dancing without music at a self-determined vigorous intensity. METs classified 83%(40) of participants as engaged in vigorous PA while dancing at a self-determined vigorous intensity with music, and 58%(28) were classified as engaged in vigorous PA while dancing at a self-determined

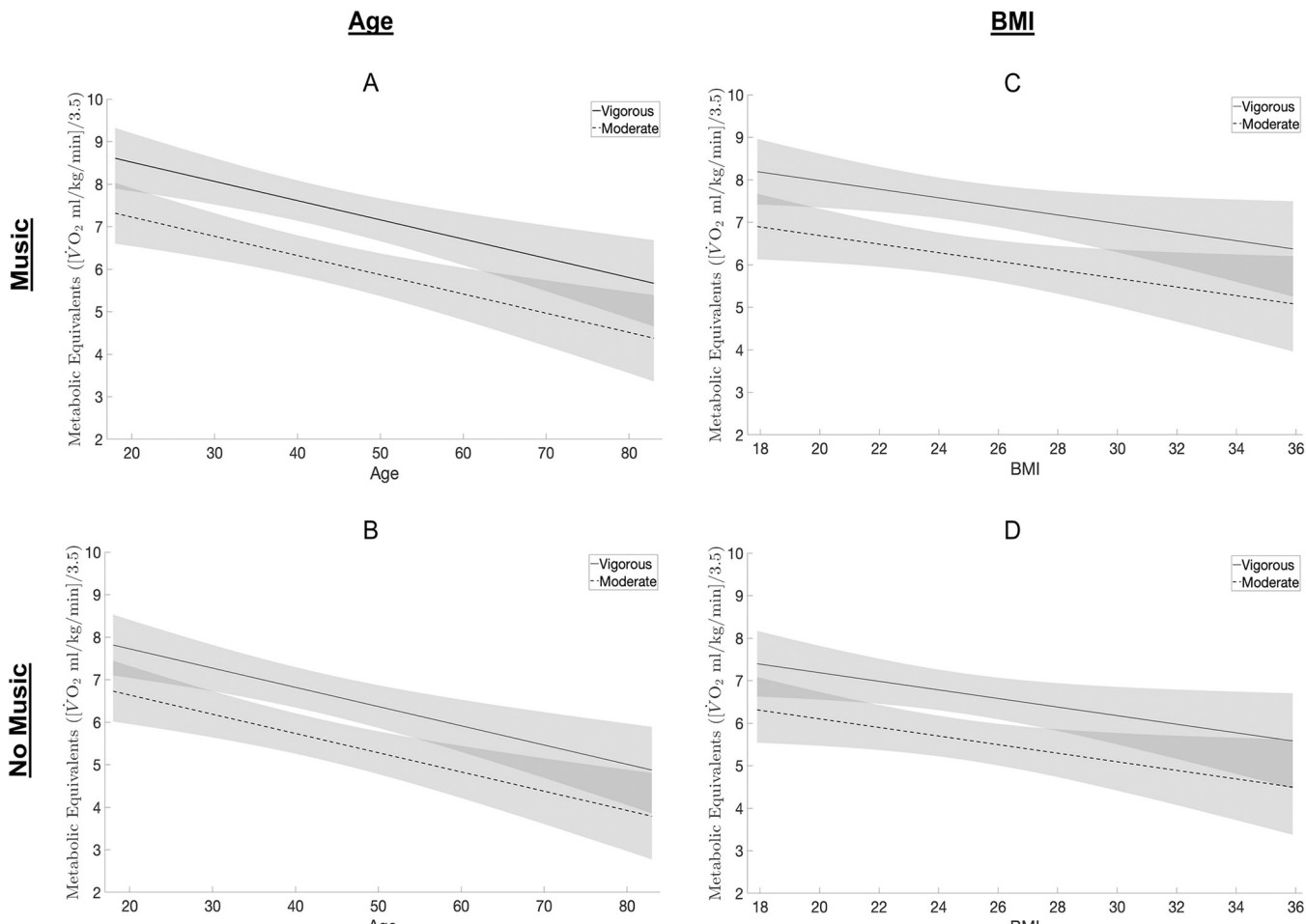

**Fig 2. Covariate-adjusted associations between the absolute intensity of free-form dance behavior, age, and BMI among 18- to 83-year-old adults** (*n* = 48). Note: Fig 2 shows regression lines and 95% confidence intervals for the linear mixed effects model on the absolute intensity (METs) of free-form dance behavior at self-determined moderate and vigorous intensities with and without music in relation to age and BMI (Model 5b). Fig 2A and 2B show a significant ($p < 0.05$) inverse relationship between METs and age, with BMI held constant at the sample mean (i.e., 24.7 kg/m$^2$). Fig 2C & 2D show a significant ($p < 0.05$) inverse relationship between METs and BMI, with age held constant at the sample mean (i.e., 42.3 years). Abbreviations: Metabolic equivalents (METs), body mass index (BMI), milliliter (ml), kilogram (kg), minute (min), meter (m).

vigorous intensity without music. The %HR$_{max}$ classified 79%(38) and 67%(32) of participants in vigorous PA during self-determined vigorous intensity dancing with music and without music, respectively.

## Discussion

Young to older adults in this study, with a range of 0 to 56 years of dance training experience, engaged in free-form dance bouts at respectively self-determined moderate and vigorous intensities with and without music. Engaging in solo, free-form dance at a self-determined moderate intensity, with or without music, was sufficient for nearly all participants to reach a moderate PA intensity as classified by METs. When classified using %HRR, all participants were engaged in MVPA when dancing at a self-determined moderate intensity with music or without music. During self-determined vigorous intensity free-form dancing, both with and without music, most participants (92% and 94%, respectively) engaged in at least a vigorous PA intensity as classified by %HRR. When classified using METs, most (>80%) participants

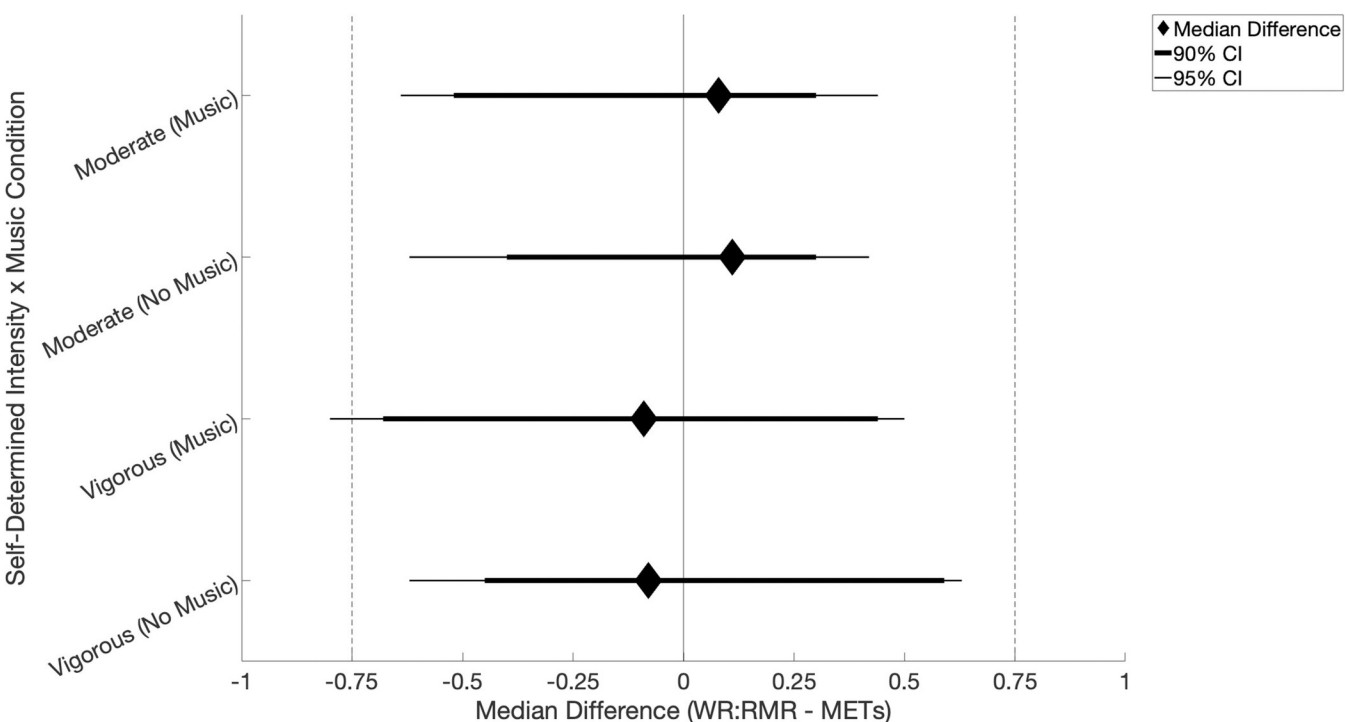

**Fig 3. Metabolic equivalents (METs) and the work rate-to-resting metabolic rate (WR:RMR) during solo, free-form dancing in the presence and absence of music.** Note: Fig 3 shows results from respective two one-sided tests of equivalence (TOST) comparing METs and the WR:RMR for each free-form dance by music condition among a cohort of 18- to 83-year-old adults ($n = 47$). Results are presented at the 95% C.I with Δ = [-0.75, 0.75], and the 90% C.I. has additionally been presented. Abbreviation: Confidence Interval (C.I.).

engaged in vigorous PA when dancing with music at a self-determined vigorous intensity, and a majority (58%) engaged in vigorous PA dancing without music. At the group level, engaging in free-form dance behavior at a self-determined moderate or vigorous intensity was sufficient for eliciting a moderate-to-vigorous PA intensity, which points toward the need for further research on the potential benefits of free-form dance as an accessible mode of health-enhancing PA. At the same time, some participants tended to perceive they were dancing at lower intensities than were otherwise observed in analyses of oxygen uptake and heart rate responses recorded during dance bouts. These divergent responses, with respect to the perceived relative intensities of free-form dancing versus the device-determined absolute and relative intensities, simultaneously suggest that further research on perceptions of effort during free-form dancing is additionally warranted.

Within this sample of 48 young to older adults, the mean absolute intensity of self-determined moderate-to-vigorous intensity solo, free-form dancing with and without music was 6.5 METs, the mean observed HR was 76% $HR_{max}$ and 74% of the HRR respectively, and the mean perceived intensity was 13 on the Borg scale. The presence of music and the intention to dance at a vigorous intensity were both positively associated with intensity for all relative intensity measures (i.e., %$HR_{max}$, %HRR, and RPE); however, there was no significant interaction between the music and intensity conditions in the sample. The presence of music and the intention to dance at a vigorous intensity were also positively associated with intensity as determined by METs, while age and BMI were inversely associated with METs during dance bouts. A 1982 study of 8 pairs of young adult disco dancers showed via V$\dot{V}$O$_2$ and HR data that dancing to disco music with a partner resulted in reaching an average 8.6 METs and

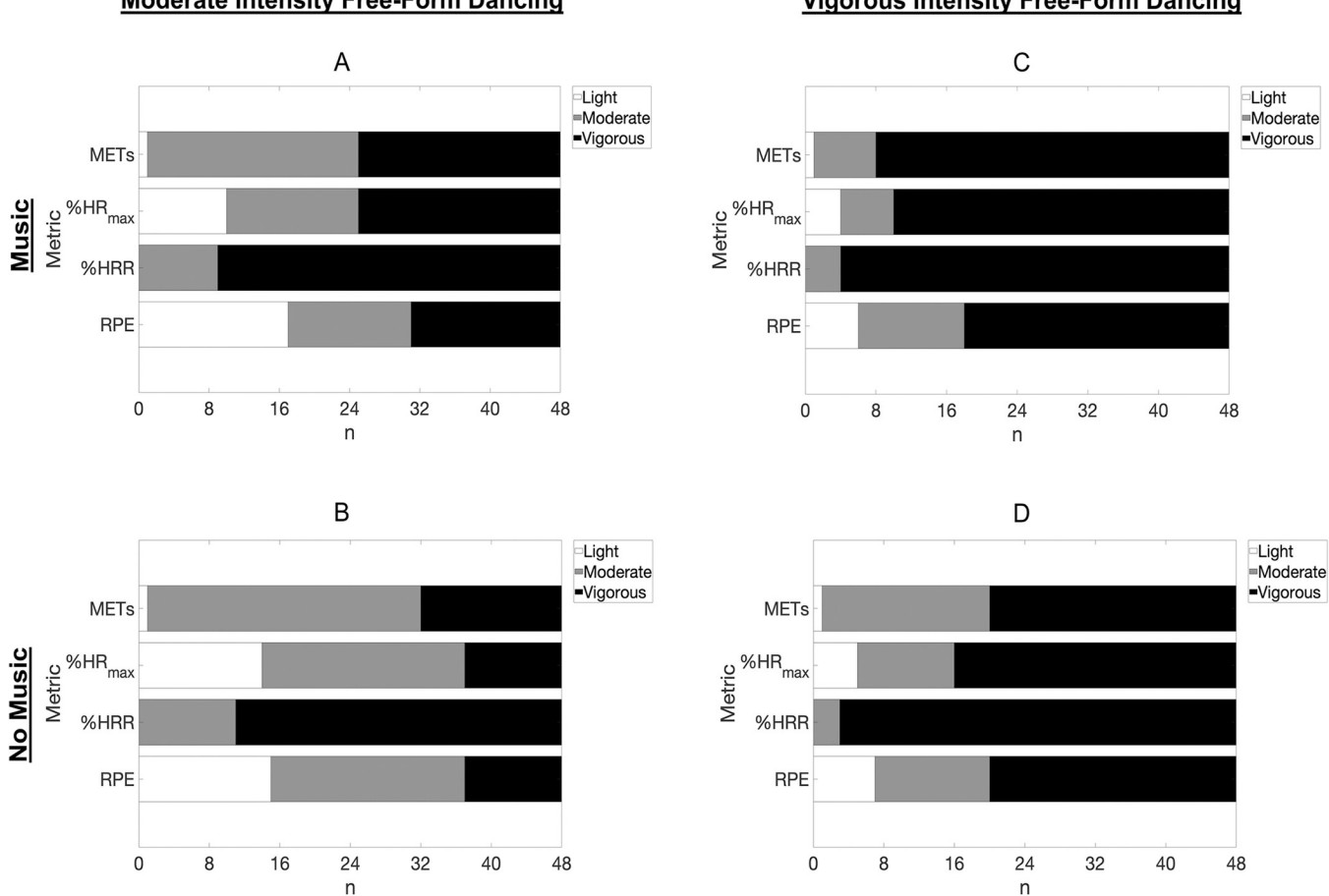

**Fig 4. Activity intensity classification for free-form dance behavior among 18- to 83-year-old adults.** Note: Absolute (METs) and relative (%HR$_{max}$, %HRR, RPE) intensities of self-determined moderate and vigorous free-form dance bouts (with and without music) classified into light, moderate, and vigorous activity intensities ($n$ = 48). Abbreviations: Metabolic equivalents (METs), percent of age-predicted maximal heart rate (%HR$_{max}$), percent of heart rate reserve (%HRR), rating of perceived exertion (RPE).

~72% of their maximal HR during steady state disco dancing [17]. Similarly, a study of 10 young adult ballroom dancers showed that engaging in ballroom dance with a partner elicited an average 85% to 91% of their HR$_{max}$ [45], and another investigation of ballroom dancing in 12 pairs of young adults observed an average of 5.3 METs to 7.1 METs via indirect calorimetry [14]. An analysis of "hambo" dancing in 6 pairs of young adults reported average absolute intensities of 10.7 METs to 11.0 METs [18]. Cohen and colleagues [15] observed in 10 professional dancers an average %HR$_{max}$ of 63% to 69% during ballet barre sequences, and indirect calorimetry results showed an average of 4.7 METs to 5.3 METs; average HR responses during center work were 74% to 79% HR$_{max}$, and METs during center work were reported as 5.7 METs to 7.5 METs. An analysis of PA intensity via indirect calorimetry during modern dance showed an average 5.0 METs to 6.3 METs in class, 2.9 METs to 4.9 METs in rehearsal, and 6.7 METs to 6.9 METs in performance among 40 young adults with prior dance training experience [16]. In all, the intensity of free-form dancing in this study was higher in the presence of music than in its absence when intensity was measured by HR$_{max}$, HRR, and METs. Thus, this study on free-form dance and PA intensity extends the existing literature on PA and music that shows the presence of familiar music during walking may be positively associated with

cadence and PA intensity [23]—or inversely associated with variability across stride velocity and stride time [21]—or that musical characteristics associated with mood labels (i.e., activating, relaxing, neutral) may have a differential influence on gait speed [20]. A study on improvisational dance showed, however, that movement qualities observed at the arms and torso did not significantly differ when compared between dance bouts that were performed in silence versus those that were accompanied by a metronome [46]. Future studies on music and dance should build upon the findings presented herein by investigating different auditory stimuli, for example the use of music versus a metronome, in relation to PA activity intensity among adults engaged in free-form dance.

Though free-form dancing is not a codified dance style, preliminary evidence from this pilot study in 48 young to older adults appears to indicate that, at the group level, absolute PA intensities elicited during self-determined moderate intensity solo, free-form dance bouts are of a similar intensity to those reported for ballroom dancing with a partner, ballet barre sequences, modern dance classes and performances, and other modes of PA like bicycling at 9.4 mph or walking backwards at 3.5 mph [13]. The absolute intensity of free-form dancing was inversely associated with age and BMI in this sample of 18- to 83-year-old adults, which is consistent with prior studies that have shown METs are associated with individual-level factors [34,47]. At the group level, self-determined vigorous intensity solo, free-form dancing appears to elicit absolute PA intensities like those reported for jogging, bicycling, walking at 4.5 mph, or playing tennis [13]; however, given that results showed METs during free-form dancing at self-determined intensities were inversely associated with age and BMI, the range of activities that are similar to free-form dancing at a self-determined intensity may vary with age and BMI when using absolute intensity to quantify dose. When using a relative measure of intensity, such as %HR$_{max}$ for example, partnered disco dancing in young adults [17] appears to elicit similar PA intensities as those observed, at the group level, within this study of self-determined moderate-to-vigorous intensity free-form dancing in 18- to 83-year-olds. Moreover, this study protocol discouraged very vigorous dancing, yet at least one participant in the pilot study did reach a PA intensity >11 METs during self-determined vigorous intensity dancing, thus indicating some healthy adults will dance at very vigorous intensities, like those seen during "hambo" dancing [18], while engaged in free-form dancing at a self-determined vigorous intensity. In this pilot study, the intention to engage in dance behavior at a self-determined moderate intensity and self-determined vigorous intensity, with or without music, were respectively sufficient stimuli for all young to older adults to reach a training intensity of >40% HRR [38] while engaging in free-form dancing. Current PA recommendations suggest that adults engage in at least 150 minutes of moderate intensity PA weekly [11]—given that HRR results in this present study showed that all adults engaged in at least a moderate PA intensity while dancing however they wished, free-form dancing at self-determined moderate or vigorous intensities appears to be a suitable mode of PA that active adults may include in their leisure time PA routine. In view of the recommendations offered in a recent umbrella review on dance and health [7] alongside results from a meta-analysis that show dancing confers especial benefits on health when compared to other PA modes [7], future studies should build upon current studies of free-form dance behavior [25,26,48,49] to investigate both the acute and cumulative training effects and health benefits associated with free-form dancing at various self-determined PA intensities. Furthermore, additional research is needed on free-form dance behavior at self-determined very vigorous intensities (>84% HR$_{max}$).

Participants perceived that they were engaged in light to vigorous PA intensities during self-determined moderate intensity free-form dancing with and without music, though $\dot{V}O_2$ and HR data recorded across bouts showed that ~23% to ~81% of the participants were

engaged in vigorous PA. During the self-determined vigorous intensity bouts completed with music, participants also perceived they were engaged in light to vigorous PA intensities, though %HRR data showed that ~92% were dancing at a vigorous PA intensity. At the same time, when asked to engage in self-determined vigorous intensity free-form dancing, participants engaged in significantly higher absolute and relative PA intensities than during their self-determined moderate intensity dance bouts. Together, these results suggest that community-dwelling young to older adults, either with or without prior dance training, are able to significantly modulate the PA intensity of their solo, free-form dancing at will, both with and without music, despite the heterogenous and idiosyncratic nature of free-form dance behavior [12]. Simultaneously, these findings importantly highlight that additional research on the use of wearable monitors and other technologies [38] for monitoring PA intensity is needed among individuals who may under- or overestimate their PA intensity while engaging in free-form dancing. Most participants in the study (~85%) required a reminder from study staff to keep their HR below 84% $HR_{max}$ while engaged in vigorous intensity dancing, and these participants all expressed surprise when told their HR had exceeded the upper intensity threshold while free-form dancing. Attention should be given to advancing research on the use of technologies that aid individuals in monitoring physiological parameters during free-form dance behavior.

As noted previously, a key delimitation in the study protocol is that participants were discouraged from dancing at very vigorous intensities (>84% $HR_{max}$), which may have led some individuals to dance less vigorously than they would have otherwise danced under unsupervised conditions. The requirement to dance at a vigorous intensity ≤84% $HR_{max}$ may also have contributed to the reported non-significant music by intensity condition interaction observed in the study, as the study staff were required to ask most participants to decrease the intensity of their dancing during at least one dance bout. A second limitation of the study is that participants in the sample were active, community-dwelling adults who were able to engage in moderate-to-vigorous intensity dancing. Additional studies are needed to establish the absolute and relative PA intensities of free-form dance behavior in population groups with health conditions that are known to impact metabolic responses to exercise. This study used the age-predicted maximal HR formula by Gellish et al., [35,36] and the use of other formulae may impact PA intensity classifications that are based on HR. Finally, a 15-breath moving average filter was applied to raw breath-by-breath oxygen uptake signals, and differences in data sampling and processing methods between studies using indirect calorimetry to measure expired gases may impact results and thereby limit precise $\dot{V}O_2$ comparisons between reports [30].

A major strength of this study is that young to older adults were invited to dance however they wanted, without music or with music of their choice; there was no requirement to learn a motor pattern from an external source. Thus, results on the absolute and relative intensities of free-form dancing reported in this study are largely generalizable to contexts in which adults may dance freely, with or without music, at self-determined moderate-to-vigorous PA intensities. To our knowledge, this is the first study to investigate PA intensities during free-form dancing in a sample of young to older adults. Additionally, measures of both the absolute and relative intensities of free-form dance behavior were reported herein in order to provide a panel of PA metrics for quantifying free-form dance exposures at self-determined moderate and vigorous PA intensities [10].

Taken together, results from this study provide evidence that engaging in free-form dancing at self-determined moderate-to-vigorous PA intensities was sufficient for most adults to reach a moderate-to-vigorous PA intensity as measured by $\dot{V}O_2$ and HR. Moreover, this study suggests that no dance specific training nor instructions are required for most adults to use solo,

free-form dancing as a mode of MVPA at will. Free-form dancing is immediately accessible to most everyone, and it is a PA mode that is unhampered by PA barriers such as neighborhood walkability. Additional research is needed to determine if active adults who under- or overestimate their PA intensity while engaged in free-form dancing would benefit from utilizing available wearable technologies [38] to provide real-time feedback while engaged in free-form dancing in order to better monitor their energy expenditure relative to their perceived level of exertion.

In conclusion, absolute and relative measures of PA intensity were used to quantify the intensity of free-form dance behavior in a sample of active, healthy adults ages 18 to 83 years old with a range of 0 to 56 years of prior dance training experience. Engaging in free-form dance behavior at a self-determined moderate-to-vigorous intensity (with or without music) was classified as MVPA in all adults by measure of %HRR. Some adults perceived they were engaged in lower or higher PA intensities than were otherwise observed in analyses of $\dot{V}O_2$ and HR data, and adults were able to volitionally increase the intensity of their free-form dancing with or without music between self-determined moderate and vigorous intensity bouts. Study results support the use of free-form dance as a mode of PA behavior that may help adults accumulate the recommended weekly dose of $\geq$150 minutes of moderate intensity PA. Future studies among adults who under- or overestimate PA intensities during free-form dancing should investigate the use of technology to help adults, when needed, better monitor energy expenditure during free-form dance bouts.

## Supporting information

**S1 Data.**
(ZIP)

## Acknowledgments

We are grateful to all participants who volunteered their time in the study. Thanks to Laboratory for the Scientific of Study of Dance team members, Kiley Baker, Brienne Donahue, Emily Loane, and Dr. Jared Ramer for their diverse support. Disclaimer: The opinions expressed are those of the author and do not represent the views of the National Endowment for the Arts (NEA) Office of Research & Analysis or the National Endowment for the Arts. The NEA does not guarantee the accuracy or completeness of the information included in this material and is not responsible for any consequences of its use.

## Author Contributions

**Conceptualization:** Aston K. McCullough.

**Data curation:** Aston K. McCullough.

**Formal analysis:** Aston K. McCullough.

**Funding acquisition:** Aston K. McCullough.

**Investigation:** Aston K. McCullough.

**Methodology:** Aston K. McCullough.

**Project administration:** Aston K. McCullough.

**Resources:** Aston K. McCullough.

**Software:** Aston K. McCullough.

**Supervision:** Aston K. McCullough.

**Validation:** Aston K. McCullough.

**Visualization:** Aston K. McCullough.

**Writing – original draft:** Aston K. McCullough.

**Writing – review & editing:** Aston K. McCullough.

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
