## [Decision Letter · Decision Letter 0]

20 May 2024

PONE-D-24-06735Absolute and relative intensities of solo, free-form dancing in adults: A pilot studyPLOS ONE

Dear Dr. McCullough,

Thank you for submitting your manuscript to PLOS ONE. After careful consideration, we feel that it has merit but does not fully meet PLOS ONE’s publication criteria as it currently stands. Therefore, we invite you to submit a revised version of the manuscript that addresses the points raised during the review process.

Please note that we have only been able to secure a single reviewer to assess your manuscript. We are issuing a decision on your manuscript at this point to prevent further delays in the evaluation of your manuscript. Please be aware that the editor who handles your revised manuscript might find it necessary to invite additional reviewers to assess this work once the revised manuscript is submitted. However, we will aim to proceed on the basis of this single review if possible. 

We look forward to receiving your revised manuscript.

Kind regards,

Avanti Dey, PhD

Staff Editor

PLOS ONE

Journal Requirements:

Additional Editor Comments (if provided):

Reviewers' comments:

Reviewer's Responses to Questions

**Comments to the Author**

1. Is the manuscript technically sound, and do the data support the conclusions?

Reviewer #1: Yes

2. Has the statistical analysis been performed appropriately and rigorously? 

Reviewer #1: Yes

3. Have the authors made all data underlying the findings in their manuscript fully available?

Reviewer #1: Yes

4. Is the manuscript presented in an intelligible fashion and written in standard English?

Reviewer #1: Yes

5. Review Comments to the Author

Reviewer #1: Overall, this manuscript presents a thorough explanation of a well-conducted study. The necessity of this study, through the evidence it aims to add to the ability of free-form dancing to meet exercise intensity criteria to enhance health, is well laid out in some places, but could be further emphasised in others – particularly in relation to the utility/application/implications of the study’s key findings.

Two general feedback points are 1) to consider including additional figures and tables. This would help to make large (and at times repetitive) portions of text in the method and results more readily understandable for the reader. Specific examples of where these might be particularly useful are outlined below. 2) throughout the manuscript, the use of acronyms is currently inconsistent, with acronyms and writing out in full used interchangeably for a number of terms. These should be defined on first use and then consistently used thereafter. Most notable is use of MVPA (where appropriate), but other examples include use of VO2/VCO2 in the measures section, and HRmax/HRR (particularly in the discussion).

Specific feedback points to consider are outlined below. The use of line numbers would have aided this review, but the feedback points provided attempt to be as directed as possible.

Abstract

- Objectives – add another sentence as to ‘why’ this is important to examine

- Method – add the specific number of dance bouts undertaken (with range if relevant) rather than stating ‘multiple…’

- Method – add the specific variable of interest to the indirect calorimetry description (i.e. calculation of METs)

- Methods – include a brief statement of the statistical analyses undertaken to give context to test statistics reported in the results section

- Results – I might suggest removing the first sentence as it is not clear why these data are included here for the moderate bouts, but not for the vigorous. I would instead suggest presenting the last two sentences of this section first, which contain summary data pertaining directly to the key finding

- Conclusions – consider adding a final sentence stating the possible implications/applications of the key findings (adding the ‘why’ element more clearly again)

Introduction

- While the focus is clearly outlined as being concerned with addressing a gap in the existing literature specifically surrounding the intensity of free-form dance activity, it would be pertinent to also consider the durational component of physical activity recommendations. E.g. what are the current recommendations for duration of sustained MVPA?

- Given the use of music as a manipulated variable within the design, it would be pertinent to also provide a summary of existing evidence on the possible influence of music (preference, tempo, volume, etc.) on self-determined exercise intensity (measured and perceived).

Methods

Sample

- Exclusion criteria are provided in detail

- Consider adding a flow diagram detailing participant numbers at each stage, i.e. number of individuals who initially responded to the open recruitment advert, number screened out based on stated exclusion criteria, and any other missing or incomplete data sets excluded from analysis

Procedures

- Refer to ‘body mass’ instead of ‘weight’ throughout

- Appropriate control measures were implemented, with the condition of participants accounted for. As per the comment above, did any participants not meet these criteria and subsequently miss measurements, have to return on a separate occasion, or need to be excluded from analysis?

- More information about the music choice process is required: how many pieces were selected? Were these from a set selection (of how many?) or completely open for the individual to choose?

- More information about the dance bouts is required: how many 5-minute bouts did they complete in total and was this consistent across participants (or a range)? Were the moderate and vigorous bouts specifically instructed (i.e. “try to reach a moderate intensity for this bout”)? If not, how were these differentiated for analysis? Were moderate bouts always performed first or were they randomised? A diagram of the experimental set up would be useful to give the reader a clearer understanding of how this was practically managed.

Measures

- Good detail is provided

- Was the participant who didn’t complete the REE measure excluded from all analyses or just these data? How was this managed? (This should be added to the sample size diagram mentioned above)

Statistical analyses

- Good detail is provided and the inclusion of justification for tests chosen is good to see

Results

- It is not clear where missing data is accounted for in the reported n (this should also be a small case n for sample size)

- Consider reporting key values in table format to allow easier comparison across conditions than is currently possible within the large amount of text presented, e.g., mean (95% CI) values for different conditions and TOST results.

Discussion

- The key findings are well discussed with reference to the exiting evidence base

- As mentioned under the introduction, the durational component for physical activity guidelines is missing here. Would the intensities seen during the five-minute bouts be sustained by participants over a longer duration? What would the mean METs and/or overall RPE be over say 20 min? This should perhaps be acknowledged as a limitation of the present study and a question to examine in future research.

6. PLOS authors have the option to publish the peer review history of their article (what does this mean?). If published, this will include your full peer review and any attached files.

Reviewer #1: **Yes: **Sarah C. Needham-Beck

---

## [Author Response · Author response to Decision Letter 0]

24 Jul 2024

Review Comments to the Author

Reviewer #1: Overall, this manuscript presents a thorough explanation of a well-conducted study. The necessity of this study, through the evidence it aims to add to the ability of free-form dancing to meet exercise intensity criteria to enhance health, is well laid out in some places, but could be further emphasised in others – particularly in relation to the utility/application/implications of the study’s key findings.

Two general feedback points are 1) to consider including additional figures and tables. This would help to make large (and at times repetitive) portions of text in the method and results more readily understandable for the reader. Specific examples of where these might be particularly useful are outlined below. 2) throughout the manuscript, the use of acronyms is currently inconsistent, with acronyms and writing out in full used interchangeably for a number of terms. These should be defined on first use and then consistently used thereafter. Most notable is use of MVPA (where appropriate), but other examples include use of VO2/VCO2 in the measures section, and HRmax/HRR (particularly in the discussion).

Specific feedback points to consider are outlined below. The use of line numbers would have aided this review, but the feedback points provided attempt to be as directed as possible.

- Thank you for the thorough review of this manuscript. The two main points above have been strongly considered, and the manuscript has been revised accordingly. Specifically, additional figures and tables have been included within the methods and results are now more visually interpretable. The re-expansion of any previously defined abbreviations (in the discussion section, specifically) has been revised, so that no abbreviations are re-expanded in the manuscript after first their initial definition. It is hoped that this revision to the abbreviations is most helpful for the readership.

Abstract

- Objectives – add another sentence as to ‘why’ this is important to examine

- Thank you. A sentence has been included that helps point toward the significance of dance as a mode of physical activity. 

- Method – add the specific number of dance bouts undertaken (with range if relevant) rather than stating ‘multiple…’

- Thank you. This sentence has been revised, and the number of bouts has been clarified within the procedures section as well.

- Method – add the specific variable of interest to the indirect calorimetry description (i.e. calculation of METs)

- Thank you. We have named and defined METs within the methods section.

- Methods – include a brief statement of the statistical analyses undertaken to give context to test statistics reported in the results section

- Yes, thank you for inviting the abstract to make space for this. While the statement about the linear mixed effects model was originally omitted to help with the word count, this important information has been returned to the manuscript.

- Results – I might suggest removing the first sentence as it is not clear why these data are included here for the moderate bouts, but not for the vigorous. I would instead suggest presenting the last two sentences of this section first, which contain summary data pertaining directly to the key finding

- Thank you. In the original submission, the first sentence referred to above was intended to ground the readers in the results by offering the intercept for the linear mixed effects models that estimated METs. The second sentence then described key independent variables in the linear mixed effects model and noted the effect of engaging in dance at a self-determined vigorous intensity. The order of the results about the absolute and relative intensities has been revised, as requested, and punction has been revised in the statements about METs to help better connect all related results. 

- Conclusions – consider adding a final sentence stating the possible implications/applications of the key findings (adding the ‘why’ element more clearly again)

- Thank you for the invitation to situate the findings more deeply into the existing literature. The conclusions have been revised to say more about the implications of the study findings—that is to say that, engaging in free-form dance can be incorporated into one’s weekly PA routine toward accumulating the recommended >=150 minutes of moderate PA.

Introduction

- While the focus is clearly outlined as being concerned with addressing a gap in the existing literature specifically surrounding the intensity of free-form dance activity, it would be pertinent to also consider the durational component of physical activity recommendations. E.g. what are the current recommendations for duration of sustained MVPA?

- Thank you for this very important public health-oriented question. The latest recommendations available from the 2018 Physical Activity Guidelines for Americans have been included as part of the revision. With this significant revision, the importance of engaging in moderate-to-vigorous physical activity bouts of any duration is cited and underscored. Thus, the significance of the study with respect to engaging in short bouts of free-form dance is further highlighted by way of responding to this important comment.

- Given the use of music as a manipulated variable within the design, it would be pertinent to also provide a summary of existing evidence on the possible influence of music (preference, tempo, volume, etc.) on self-determined exercise intensity (measured and perceived).

- The introduction has been revised to further highlight the significance of music within the study design.

Methods

Sample

- Exclusion criteria are provided in detail

- Thank you.

- Consider adding a flow diagram detailing participant numbers at each stage, i.e. number of individuals who initially responded to the open recruitment advert, number screened out based on stated exclusion criteria, and any other missing or incomplete data sets excluded from analysis

- Thank you for this suggestion. The methods section now provides detailed descriptions of the number of participants who provided informed consent, in addition to the number that withdrew from the study, along with reasons for participant withdrawal from the study. In the original manuscript, focus was placed exclusively on the participants who returned to complete the free-form dance session; in the revision a detailed description has been interleaved into the text to describe the participant flow through the study activities after obtaining informed consent.

Procedures

- Refer to ‘body mass’ instead of ‘weight’ throughout

- Thank you. The manuscript has been updated to use the term ‘body mass’

o doi: 10.1097/NT.0000000000000092

- Appropriate control measures were implemented, with the condition of participants accounted for. As per the comment above, did any participants not meet these criteria and subsequently miss measurements, have to return on a separate occasion, or need to be excluded from analysis?

- Please see response to comment above.

- More information about the music choice process is required: how many pieces were selected? Were these from a set selection (of how many?) or completely open for the individual to choose?

- Thank you very much. Importantly, this revision clarifies that it was completely open for every individual to choose each piece of music from their own personal music library.

- More information about the dance bouts is required: how many 5-minute bouts did they complete in total and was this consistent across participants (or a range)? Were the moderate and vigorous bouts specifically instructed (i.e. “try to reach a moderate intensity for this bout”)? If not, how were these differentiated for analysis? Were moderate bouts always performed first or were they randomised? A diagram of the experimental set up would be useful to give the reader a clearer understanding of how this was practically managed.

- Thank you; a diagram has been included as suggested (Figure 1). We have also clarified that the self-determined intensities across moderate and vigorous intensity bouts were respectively determined by each individual participant. The number of bouts has also been underscored as being four. 

Measures

- Good detail is provided

- Thank you.

- Was the participant who didn’t complete the REE measure excluded from all analyses or just these data? How was this managed? (This should be added to the sample size diagram mentioned above)

- The participant without REE was not included in reports for any test that used REE (i.e., WR:RMR and descriptive statistics of REE). 

- The original version of the manuscript underscored the sample size in each figure and table for the readers. The lower sample size for REE and WR:RMR were respectively indicated in the original manuscript within the methods and results section. 

- Now, in responding to the suggestion to translate key statistical results into a table, the sample size has been underscored across each place results are reported to ensure readers can confirm the sample size for each test. 

- Therefore, the associated sample size is clearly presented to the readership for every statistical test that was conducted.

Statistical analyses

- Good detail is provided and the inclusion of justification for tests chosen is good to see

- Thank you for your careful review. 

Results

- It is not clear where missing data is accounted for in the reported n (this should also be a small case n for sample size)

- Thank you for identifying the secondary capitalized ‘N’ typo -- the case of the sample size indicator (n) has been appropriately revised. 

- Please note response to question above regarding sample size and missing cases

- Consider reporting key values in table format to allow easier comparison across conditions than is currently possible within the large amount of text presented, e.g., mean (95% CI) values for different conditions and TOST results.

- Thank you for inviting this revision to include additional key summary tables and figures. The revision now includes a summary table of the absolute and relative intensities for each condition, along with 95% CI. The revision also includes a typical TOST figure that allows for visual comparison of the 95% CI across conditions.

Discussion

- The key findings are well discussed with reference to the exiting evidence base

- Thank you.

- As mentioned under the introduction, the durational component for physical activity guidelines is missing here. Would the intensities seen during the five-minute bouts be sustained by participants over a longer duration? What would the mean METs and/or overall RPE be over say 20 min? This should perhaps be acknowledged as a limitation of the present study and a question to examine in future research.

- Thank you again for this comment that ties the introduction and discussion sections to the current PA recommendations for adults. The revised manuscript discusses the current recommendation that >=150 minutes of moderate intensity PA should be accumulated weekly in bouts of any duration, per the latest recommendations presented in the 2018 Physical Activity Guidelines for Americans.

---

## [Decision Letter · Decision Letter 1]

27 Aug 2024

PONE-D-24-06735R1Absolute and relative intensities of solo, free-form dancing in adults: A pilot studyPLOS ONE

Dear Dr. McCullough,

Thank you for submitting your manuscript to PLOS ONE. After careful consideration, we feel that it has merit but does not fully meet PLOS ONE’s publication criteria as it currently stands. Therefore, we invite you to submit a revised version of the manuscript that addresses the points raised during the review process.

We look forward to receiving your revised manuscript.

Kind regards,

Maja Vukadinovic

Academic Editor

PLOS ONE

Journal Requirements:

Reviewers' comments:

Reviewer's Responses to Questions

**Comments to the Author**

1. If the authors have adequately addressed your comments raised in a previous round of review and you feel that this manuscript is now acceptable for publication, you may indicate that here to bypass the “Comments to the Author” section, enter your conflict of interest statement in the “Confidential to Editor” section, and submit your "Accept" recommendation.

Reviewer #1: All comments have been addressed

Reviewer #2: (No Response)

2. Is the manuscript technically sound, and do the data support the conclusions?

Reviewer #1: Yes

Reviewer #2: Yes

3. Has the statistical analysis been performed appropriately and rigorously? 

Reviewer #1: Yes

Reviewer #2: Yes

4. Have the authors made all data underlying the findings in their manuscript fully available?

Reviewer #1: Yes

Reviewer #2: Yes

5. Is the manuscript presented in an intelligible fashion and written in standard English?

Reviewer #1: Yes

Reviewer #2: Yes

6. Review Comments to the Author

Reviewer #1: Thank you for the detail paid in responding to comments and submitting the revised version of the manuscript. I believe this is a detailed account of a well conducted and technically sound study, that is suitable for publication.

Reviewer #2: The Manuscript presents a pilot study characterizing the absolute and relative physical activity (PA) intensities of solo, free-form dancing at self-determined moderate and vigorous intensities in adults. The examination was set to investigate if dancing however one wishes at such perceived intensities may support adults in accumulating the recommended weekly dose of equal or more than 150 minutes of moderate intensity PA. The BMI and use of music background for activity were considered.

The Manuscript provided data that supports the conclusions: thorough presentation of data collection and analysis.

The research was conducted rigorously, with appropriate controls. The procedures were planned with great care in a detailed manner including the criteria determining the choice of participants of the examined group as well as conditions in which examination took place. Absolute intensity was measured during free-form dancing bouts using indirect calorimetry (metabolic equivalents; METs = V˙ O2 ml·kg-1·min-1/3.5). Relative intensity was measured by ratings of perceived exertion (Borg scale) and heart rate. Linear mixed effects models were used assess the relationship between absolute and relative intensity metrics and model covariates. Body Mass Index (BMI) was measured using a calibrated Seca 286 stadiometer and scale (Seca, Hamburg, Germany). Sociodemographic characteristics of the group were collected by self-report questionnaires administered to participants via REDCap – they also included self-reported data of total years of dance training experience. Music Information Retrieval Toolbox was used to extract the tempo of each self-selected musical piece that participants chose to accompany their dance bouts.

Ethical approval for the study procedures and methods was granted. The human subjects research protocol conducted in the study was in adherence with the policies and procedures of, and approved by, the Institutional Review Board at the University of Massachusetts Amherst (Protocol #2070). Written and verbal consent was gained from participants prior to study participation.

The conclusions were drawn appropriately based on the data presented and clearly referred to the hypotheses formulated at the beginning of the research process. The statistical analysis was performed appropriately and rigorously. Data were analyzed in MATLAB (The Mathworks Inc. MATLAB R2022a.). To test for equivalence between music tempi in the self-selected moderate and vigorous intensity bouts, a non-parametric two one-sided tests of equivalence (TOST) model was implemented. Linear mixed effects models (Snijders, T. A. B. & Bosker, R. J., 2012) were used to estimate associations between the relative and absolute intensity of free-form dance bouts, using each respective measure of physical activity intensity, self-determined intensity (intensity condition), and sociodemographic (age), anthropometric (BMI), and environmental factors (presence of music).

The authors successfully proved that adults were able to modulate the PA intensity of free-form dancing at will between self-determined moderate and vigorous intensity dance bouts. The presence of music was positively associated with the absolute PA intensity of free-form dancing. When characterized using HRR, engaging in free-form dance at self-determined moderate-to-vigorous PA intensities provided a sufficient stimulus for all adults to reach a moderate PA intensity, which provides evidence that dancing however one wishes at such perceived intensities may support adults in accumulating the recommended weekly dose of equal or more than 150 minutes of moderate intensity PA.

All data underlying the findings in The Manuscript is fully available. It can be found within the manuscript and its Supporting Information files.

The Manuscript is presented in an intelligible fashion and written in standard English with a good proficiency.

Comments:

- “Dance” or “Free Dance” would be suggested to add in the Key Word section, as it is main activity examined in the study.

- There is some inconsistency in usage of letter “s” or “z” in the word “analyze”. Unification throughout The Manuscript would be advised. There are also minor typos to correct, for example: “tempi” instead “tempo”.

- The tables and charts could be formatted to fit into text.

- Since The Manuscript refers to “Free Dance” and “Improvisation”, extended characteristics of these activities in the theoretical introduction would be advised

- Although The Manuscript presents a pilot study, small control group could be considered for reference.

7. PLOS authors have the option to publish the peer review history of their article (what does this mean?). If published, this will include your full peer review and any attached files.

Reviewer #1: **Yes: **Sarah Needham-Beck

Reviewer #2: **Yes: **Paulina Wycichowska

---

## [Author Response · Author response to Decision Letter 1]

20 Sep 2024

Review Comments to the Author 

Reviewer #1: Thank you for the detail paid in responding to comments and submitting the revised version of the manuscript. I believe this is a detailed account of a well conducted and technically sound study, that is suitable for publication.  

Many thanks again for your thoughtful review of this manuscript. The revisions that were suggested strengthened the manuscript.  

Reviewer #2: The Manuscript presents a pilot study characterizing the absolute and relative physical activity (PA) intensities of solo, free-form dancing at self-determined moderate and vigorous intensities in adults. The examination was set to investigate if dancing however one wishes at such perceived intensities may support adults in accumulating the recommended weekly dose of equal or more than 150 minutes of moderate intensity PA. The BMI and use of music background for activity were considered.  

The Manuscript provided data that supports the conclusions: thorough presentation of data collection and analysis.  

The research was conducted rigorously, with appropriate controls. The procedures were planned with great care in a detailed manner including the criteria determining the choice of participants of the examined group as well as conditions in which examination took place. Absolute intensity was measured during free-form dancing bouts using indirect calorimetry (metabolic equivalents; METs = V˙ O2 ml·kg-1·min-1/3.5). Relative intensity was measured by ratings of perceived exertion (Borg scale) and heart rate. Linear mixed effects models were used assess the relationship between absolute and relative intensity metrics and model covariates. Body Mass Index (BMI) was measured using a calibrated Seca 286 stadiometer and scale (Seca, Hamburg, Germany). Sociodemographic characteristics of the group were collected by self-report questionnaires administered to participants via REDCap – they also included self-reported data of total years of dance training experience. Music Information Retrieval Toolbox was used to extract the tempo of each self-selected musical piece that participants chose to accompany their dance bouts.  

Ethical approval for the study procedures and methods was granted. The human subjects research protocol conducted in the study was in adherence with the policies and procedures of, and approved by, the Institutional Review Board at the University of Massachusetts Amherst (Protocol #2070). Written and verbal consent was gained from participants prior to study participation.  

The conclusions were drawn appropriately based on the data presented and clearly referred to the hypotheses formulated at the beginning of the research process. The statistical analysis was performed appropriately and rigorously. Data were analyzed in MATLAB (The Mathworks Inc. MATLAB R2022a.). To test for equivalence between music tempi in the self-selected moderate and vigorous intensity bouts, a non-parametric two one-sided tests of equivalence (TOST) model was implemented. Linear mixed effects models (Snijders, T. A. B. & Bosker, R. J., 2012) were used to estimate associations between the relative and absolute intensity of free-form dance bouts, using each respective measure of physical activity intensity, self-determined intensity (intensity condition), and sociodemographic (age), anthropometric (BMI), and environmental factors (presence of music).  

The authors successfully proved that adults were able to modulate the PA intensity of free-form dancing at will between self-determined moderate and vigorous intensity dance bouts. The presence of music was positively associated with the absolute PA intensity of free-form dancing. When characterized using HRR, engaging in free-form dance at self-determined moderate-to-vigorous PA intensities provided a sufficient stimulus for all adults to reach a moderate PA intensity, which provides evidence that dancing however one wishes at such perceived intensities may support adults in accumulating the recommended weekly dose of equal or more than 150 minutes of moderate intensity PA.  

All data underlying the findings in The Manuscript is fully available. It can be found within the manuscript and its Supporting Information files.  

The Manuscript is presented in an intelligible fashion and written in standard English with a good proficiency.  

Thank you kindly for your rigorous review of the manuscript. The specificity and generous attention to detail are greatly appreciated.  

Comments:  

- “Dance” or “Free Dance” would be suggested to add in the Key Word section, as it is main activity examined in the study.  

Thank you very much for this suggestion. Dance has been included as a keyword.  

There is some inconsistency in usage of letter “s” or “z” in the word “analyze”. Unification throughout The Manuscript would be advised. There are also minor typos to correct, for example: “tempi” instead “tempo”.  

Thank you for the suggestion to unify the style of spelling with reference to “analyses” (noun, plural) and “analyze” (verb). In the original manuscript “analyze” (verb) was used (U.S. English) and “analyses” (noun, plural) was used, which appear to be unified for this (U.S. English) reader. 

The manuscript has been carefully reviewed for any possible typos, and where tempi is used the plural form of tempo is intended.   

- The tables and charts could be formatted to fit into text.  

We believe that here the reviewer is kindly suggesting that the revision should clearly indicate the intended placement of tables and figures in the text. We have made updates throughout the manuscript to indicate how we envisioned the tables/figures might possibly be formatted to fit into the final text.  Thank you. 

- Since The Manuscript refers to “Free Dance” and “Improvisation”, extended characteristics of these activities in the theoretical introduction would be advised  

In the original manuscript, free-form dance was operationalized by inviting participants to dance as they wanted. In this revised manuscript, the introduction now expands on statements from the prior version to help connect the theoretical introduction to the procedures detailed in the methods. 

Introduction (pg. 3): 

“To dance as one wants, or feels inspired at any moment, is fundamental to engaging in free-form dance behavior—ostensibly, one can choose to dance however one wishes, most wherever one wants, and with few resources.” 

“Free-form dance behavior, in contrast to codified dance styles, is highly heterogenous in its expression given the individualized, idiosyncratic characteristics of improvisational dancing and its general lack of rehearsed motor sequences.” 

Methods (pg. 5) 

“Participants were invited to “dance in their usual fashion” both with and without music for five-minute bouts at respectively self-determined moderate and vigorous activity intensities (see Figure 1).” 

“Before participants began dancing, the researcher informed each participant that they ‘would not really be watching’ the participant dance and that they would be ‘mostly focused on the data’ during the dance bouts.” 

The theoretical introduction leverages the available, relevant peer-reviewed literature without including the opinion of the author to characterize free-form dance; additional research is urgently needed on the kinematic characteristics of free-form dance. 

Thank you for this question that further reveals the gap in the literature which this study seeks to begin to fill—that is, there is a dearth of peer-reviewed research that characterizes free-form dance to date. With the hopeful publication of this paper, it will be the first in a series of manuscripts that seek to comprehensively investigate free-form dance. 

- Although The Manuscript presents a pilot study, small control group could be considered for reference.  

Thank you for this important point. In writing this paper, the manuscript's length and content became substantial and complete. Data on self-paced walking (a control condition) in comparison to free-form dancing were saved for further analysis and future publication, which will offer an opportunity to report a detailed description of those methods, results, and further discussion. The data on WR:RMR also use a resting measure (RMR) as a reference against which PA intensity during dance was normalized.

---

## [Decision Letter · Decision Letter 2]

21 Oct 2024

Absolute and relative intensities of solo, free-form dancing in adults: A pilot study

PONE-D-24-06735R2

Dear Dr. McCullough,

We’re pleased to inform you that your manuscript has been judged scientifically suitable for publication and will be formally accepted for publication once it meets all outstanding technical requirements.

Kind regards,

Maja Vukadinovic

Academic Editor

PLOS ONE

Additional Editor Comments (optional):

Reviewers' comments:

Reviewer's Responses to Questions

**Comments to the Author**

1. If the authors have adequately addressed your comments raised in a previous round of review and you feel that this manuscript is now acceptable for publication, you may indicate that here to bypass the “Comments to the Author” section, enter your conflict of interest statement in the “Confidential to Editor” section, and submit your "Accept" recommendation.

Reviewer #2: All comments have been addressed

2. Is the manuscript technically sound, and do the data support the conclusions?

Reviewer #2: Yes

3. Has the statistical analysis been performed appropriately and rigorously? 

Reviewer #2: I Don't Know

4. Have the authors made all data underlying the findings in their manuscript fully available?

Reviewer #2: Yes

5. Is the manuscript presented in an intelligible fashion and written in standard English?

Reviewer #2: Yes

6. Review Comments to the Author

Reviewer #2: (No Response)

7. PLOS authors have the option to publish the peer review history of their article (what does this mean?). If published, this will include your full peer review and any attached files.

Reviewer #2: **Yes: **Paulina Wycichowska

---

## [Editor Report · Acceptance letter]

8 Nov 2024

PONE-D-24-06735R2 

PLOS ONE

Dear Dr. McCullough, 

I'm pleased to inform you that your manuscript has been deemed suitable for publication in PLOS ONE. Congratulations! Your manuscript is now being handed over to our production team.

Kind regards, 

on behalf of

Dr. Maja Vukadinovic 

Academic Editor

PLOS ONE